# Bioreactance assessment of cardiac output lacks reliability for the follow-up of patients with pulmonary hypertension

Ségolène Turquier[1,2]*, Laure Huot[3,4], Medhi Lamkhioued[1,2], Fabien Subtil[5], Julie Traclet[1], Kais Ahmad[1], François Lestelle[1], Louis Chauvelot[1], Vincent Cottin[1,6], Jean-François Mornex[1,6]

1 National Reference Centre for Rare Pulmonary Diseases and Centre for Pulmonary Hypertension, Louis Pradel Hospital, Hospices Civils de Lyon, Lyon, France, 2 Lung Physiology Unit, Louis Pradel Hospital, Hospices Civils de Lyon, Lyon, France, 3 Innovation Department, Health Economic Evaluation Service, Public Health Centre, Hospices Civils de Lyon, Lyon, France, 4 Research on Healthcare Performance RESHAPE, INSERM U1290, Claude Bernard University, Lyon, France, 5 Biostatistics and Bioinformatics Department, Public Health Centre, Hospices Civils de Lyon, Lyon, France, 6 UMR754, INRAE, Claude Bernard University, Lyon, France

* segolene.turquier@chu-lyon.fr

**Data Availability Statement:** All relevant data are within the manuscript.

**Funding:** The author(s) received no specific funding for this work.

## Abstract

Cardiac output (CO) is one of the primary prognostic factors evaluated during the follow-up of patients treated for pulmonary hypertension (PH). It is recommended that it be measured using the thermodilution technique during right heart catheterization. The difficulty to perform iterative invasive measurements on the same individual led us to consider a non-invasive option. The aims of the present study were to assess the agreement between CO values obtained using bioreactance (Starling™ SV) and thermodilution, and to evaluate the ability of the bioreactance monitor to detect patients whose CO decreased by more than 15% during follow-up and, accordingly, its usefulness for patient monitoring. A prospective cohort study evaluating the performance of the Starling™ SV monitor was conducted in patients with clinically stable PH. Sixty patients referred for hemodynamic assessment were included. CO was measured using both the thermodilution technique and bioreactance during two follow-up visits. A total of 60 PH patients were included. All datasets were available at the baseline visit (V0) and 50 of them were usable during the follow-up visit (V1). Median [IQR] CO was 4.20 l/min [3.60–4.70] when assessed by bioreactance, and 5.30 l/min [4.57–6.20] by thermodilution (p<0.001). The Spearman correlation coefficient was 0.51 [0.36–0.64], and the average deviation on Bland-Altman plot was -1.25 l/min (95% CI [-1.48–1.01], p<0.001). The ability of the monitor to detect a variation in CO of more than 15% between two follow-up measurements, when such variation existed using thermodilution, was insufficient for clinical practice (AUC = 0.54, 95% CI [0.33–0.75]).

## Introduction

Pulmonary hypertension (PH) is a rare disease characterized by pulmonary vascular remodeling. Its natural history is marked by a progressive decline in cardiac output (CO), which

**Competing interests:** The authors have declared that no competing interests exist.

ultimately leads to right ventricular failure and death. Risk-stratification tools used for patient monitoring include hemodynamic variables [1]. The cardiac index (CI) and the stroke volume index (SVI) are parameters derived from the CO: CI = CO/body surface, SVI = CI/heart rate. They are very strong predictors of outcomes and, therefore, it could be of interest to regularly reevaluate them during the course of the disease [1–3]. Measurement of CO by thermodilution requires right heart catheterization, which is an invasive procedure, associated with a morbidity rate between 0.9% and 1.1% and a mortality rate of less than 0.055%, when performed in experienced centers [4, 5].

Starling™ SV (CHEETAH Medical Inc, Wilmington, DE, USA) is a non-invasive CO monitoring technique, based on thoracic bioreactance technology. Previous studies evaluated the performance of this monitor at rest and during exercise [6–9]. Others explored its application in the management of intensive care, perioperative and neonatal patients. However, the results of these different studies are contradictory [10–17]. To date, only one study was conducted in a cohort of 50 patients with PH. Rich *et al.* found that bioreactance accurately measures CO at rest and reliably evaluates changes in CO after an acute vasodilator test [18].

The aims of the present study were thus to assess the correlation between CO values obtained using bioreactance (Starling™ SV) and thermodilution as a gold standard, and to evaluate the ability of the bioreactance monitor to detect patients' worsening.

## Materials and methods

### Study design and patients

This study was a single-center prospective cohort conducted in the Louis Pradel hospital (Hospices civils de Lyon, France), a French PH expert center, between May 2019 and May 2022. To be included, patients had to be over 18 years old, had precapillary PH (clinical classification groups 1,4,5) without associated cardiac or respiratory diseases (clinical classification groups 2 or 3), and no intracardiac shunts or tricuspid regurgitation. Reliable and reproducible CO measurements by thermodilution were also required, with a CI $\geq$ 2.5l/min/m$^2$ and considered stable when compared to the previous visit.

This study was approved by a French Ethics Committee (*Comité de Protection des Personnes CPP Sud Méditerranée V*) in March 2019 and was registered on clinicaltrials.gov (NCT03890627). All patients were provided with a written information and expressed their non-objection to participate in this study before inclusion, in accordance with French law.

### Outcome measures

CO measurements by thermodilution and bioreactance were performed at baseline (V0) and at the next follow-up visit (V1), expected to occur at 12 ± 6 months.

The same clinical investigator performed all procedures. For right heart catheterization, venous access was achieved with a 6-French Swan-Ganz catheter (Edwards Lifesciences, Irvine, CA, USA) inserted into the humeral vein. After pressure measurements, CO was assessed using the thermodilution technique (mean values calculated from three measurements having less than a 10% difference). Immediately after right heart catheterization, non-invasive bioreactance measurements were performed using the Starling™ SV (CHEETAH Medical Inc, distributed in France by the SEBAC laboratory). Patients were in the same position, i.e. motionless and lying on their back. Four sensor patches were placed on their thorax (upper right and left sides, lower right and left sides). Each patch consisted of a double electrode that emitted and recorded a high-frequency current. The phase shift between the upper and the lower electrodes is proportional to the aortic flow and simultaneously measures two

CO values every minute (right and left). The average of 10 measurements was collected at each visit.

Patient' worsening was defined as a CO decreased by more than 15% using thermodilution measure between baseline and the follow-up visit.

### Statistical analysis

To ensure 90% statistical power, CO measurements had to be performed on a sample size of 60 consecutive patients. Continuous variables were expressed as median (interquartile range, IQR) and categorical variables were expressed as counts and percentages of different modalities. The correlations between the two techniques (thermodilution and bioreactance) were determined using the Spearman correlation coefficient. The agreement between the two techniques was assessed using a Bland-Altman plot. The mean bias was calculated along with the 95% limits of agreement [19]. The ability of the variation in CO measurement by bioreactance to distinguish stable patients from those worsening between the two visits, compared with that of thermodilution, was evaluated using the area under the Receiver Operating Characteristic (ROC) curve and the associated 95% confidence interval (95%CI). Data were analyzed using R software version 4.0.2.

## Results

### Characteristics of the patients

A total of 60 clinically stable PH patients were included. Clinical and hemodynamic characteristics are reported in Table 1.

In accordance with the last clinical classification of PH, 75% (n = 45/60) of cases were attributed to group 1 (pulmonary arterial hypertension) [1]. Among these, the distribution was as follows: 31.7% idiopathic; 5.0% heritable; 6.7% associated with drug; 15% associated with connective tissue disease; 3.3% associated with operated congenital heart disease; 1.7% associated with human immunodeficiency virus infection; 8.3% associated with portal hypertension; 3.3% pulmonary veno-occlusive disease. A total of 20% (n = 12/60) of patients were in group 4 (PH associated with pulmonary artery obstructions) and 5% (n = 3/60) in group 5 (PH with unclear and/or multifactorial mechanisms). The median delay between diagnosis and inclusion was 3.4 years [1.4–6.3]. PH was mild to moderate, with preserved median CI (3.2 l/min/m$^2$ [2.7–3.4]) and a moderate increased of pulmonary vascular resistance (PVR: 4.3 wood units [3.2–5.2]).

### Correlation analysis

The median time between the two visits was 14.7 months [14.0–16.4]. Nine patients were withdrawn due to intercurrent events between the two visits (i.e. four death, one lung transplantation, and four patients infected with COVID-19). CO measurement by thermodilution during V1 proved unreproducible for one patient, whose data were not included in the analysis. Bioreactance made it easy to measure CO in 15 minutes at the patient's bedside. No patient experienced any intolerance or allergy to the patches.

A total of 110 datasets from 60 patients were analyzed (Table 2). The median CO was 4.20 l/min [3.60–4.70] when evaluated using bioreactance (CO BioR), and 5.30 l/min [4.57–6.20] when measured using thermodilution (CO TD). CO TD was statistically higher than CO BioR ($p < 0.001$, Fig 1).

A moderate correlation was found between CO BioR and CO TD (r = 0.51 [0.36, 0.4], $p < 0.001$, Fig 2). The average deviation on the Bland-Altman plot was -1.25l/min, with 95%

**Table 1. Patients clinical and hemodynamic characteristics at baseline.**

| | |
|---|---|
| Patients, n | 60 |
| Age, years | 66.5 [53.8–73.2] |
| Sex, female, | 41 (68.3) |
| Body mass index, Kg/m$^2$ | 24.9 [22.6–29.3] |
| PH clinical classification group, | |
| Group 1 | 45 (75) |
| Group 4 | 12 (20) |
| Group 5 | 3 (5) |
| PH specific treatment, | |
| Endothelin receptor antagonists | 45 (75) |
| Phosphodiesterase-5 inhibitors | 44 (73.3) |
| Soluble guanylate cyclase stimulator | 6 (10) |
| Prostanoid | 22 (36.7) |
| WHO functional class, | |
| I | 18 (30.0) |
| II | 35 (58.3) |
| III | 6 (10.0) |
| IV | 1 (1.7) |
| 6MWD, m | 428 [377–514] |
| BNP, ng/L | 41.5 [16–84] |
| Heart rate, /min | 72 [64–77] |
| Hemodynamic characteristics | |
| RAP, mmHg | 5 [4–7] |
| mPAP, mmHg | 31.5 [27–37.5] |
| PAWP, mmHg | 9 [7–11] |
| PVR, Wood Units | 4.3 [3.2–5.2] |
| SVO$^2$, % | 68 [65–71] |
| CI TD, l/min/m$^3$ | 3.2 [2.7–3.4] |

Data are expressed as median [interquartile range], or n (%)

PH: pulmonary hypertension, WHO: World Health Organization, 6MWD: 6-minute walk distance, BNP: brain natriuretic peptide, RAP: right atrial pressure, mPAP: mean pulmonary artery pressure, PAWP: pulmonary arterial wedge pressure, SVO$^2$: Mixed venous oxygen saturation, CI TD: cardiac index obtained by thermodilution, PVR: pulmonary vascular resistance.

**Table 2. Cardiac output obtained by thermodilution and bioreactance.**

| Patients, n | Visit 0 | Visit 1 | Total |
|---|---|---|---|
| | 60 | 50 | 110 |
| CO TD, l/min | 5.30 [4.60–5.98] | 5.20 [4.32–6.35] | 5.30 [4.57–6.20] |
| CO BioR, l/min | 4.00 [3.60–4.60] | 4.20 [3.52–4.70] | 4.20 [3.60–4.70] |

Data are expressed as median [interquartile range]

CO TD: cardiac output obtained by thermodilution

CO BioR: cardiac output obtained by bioreactance

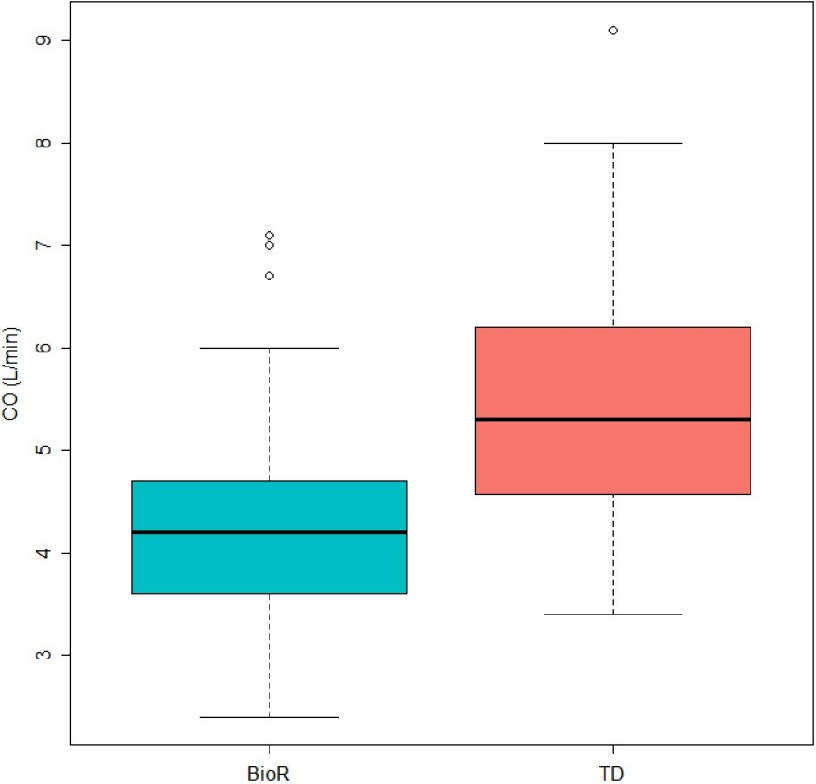

**Fig 1. Comparison of CO measurements at rest between bioreactance (BioR) and thermodilution (TD) methods.**
Median CO BioR: 4.20 l/min [Q1-Q3: 3.60–4.70], Median CO TD 5.30 l/min [Q1-Q3: 4.57–6.20] ($p<0.001$).

limits of agreement of [-1.48,-1.01]. The accuracy of bioreactance worsened as the degree of CO increased (Fig 3).

## Performance of bioreactance to detect worsening CO

Evolution over time was assessed in 50 patients. In 10 patients (20% of the cohort), CO measured using thermodilution decreased by more than 15%, while it remained stable for the other 40 patients. The ability of the bioreactance monitor to detect CO worsening, when it occurred according to thermodilution, was poor, with an area under the ROC curve (AUC) was 0.54 [95% CI: 0.33, 0.75] (Fig 4).

## Discussion

The present study showed a poor agreement between bioreactance and thermodilution techniques regarding CO measurement in patients with PH. CO measured by bioreactance was underestimated by an average of 20% compared with that measured by thermodilution. This difference between the two techniques was more pronounced as the CO increased.

To date, only one study was conducted to assess the performance of CO measurement using bioreactance in patients with PH. Rich *et al.* compared CO measurements obtained by bioreactance with those obtained by thermodilution and indirect Fick methods in 50 patients [18]. They found no statistical difference between the CO values obtained by bioreactance and indirect Fick (BioR 4.73±1.15 l/min vs Fick 4.84 ± 1.39 l/min respectively, p = 0.58), which led

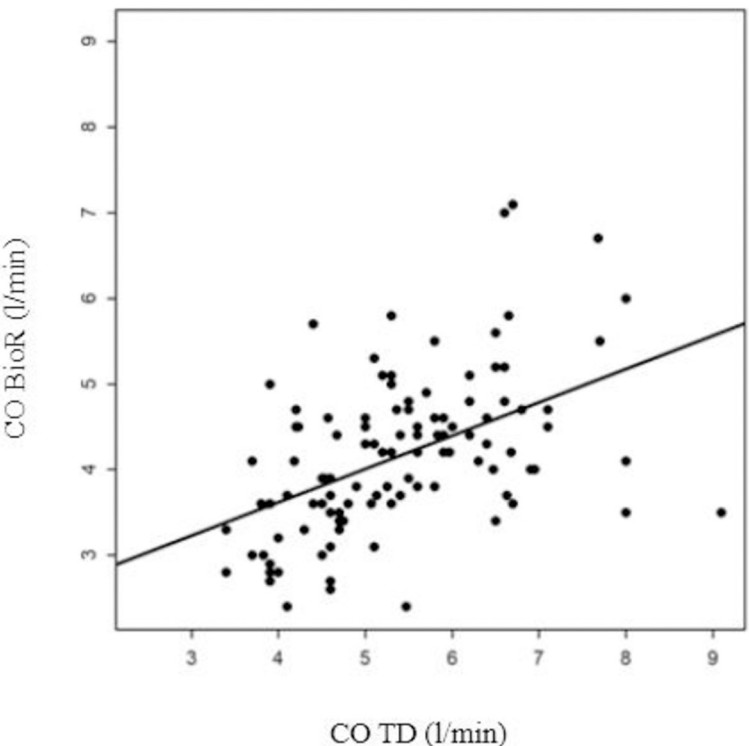

**Fig 2. Linear regression analysis using Spearman correlation between CO measured by bioreactance (CO BioR) and CO thermodilution (r = 0.51[0.36–0.64]; $p$<0.001).**

the authors to conclude that bioreactance may allow non-invasive hemodynamic assessment of patients with PH. However, the values obtained by bioreactance and thermodilution were, as herein, statistically different (BioR 4.73±1.15 l/min vs TD 5.69 ± 1.74 l/min respectively, p<0.01). The Bland-Altman analysis was also disappointing, as the average deviation was -0.81 [-3.54–1.92]. International guidelines specify that CO should be assessed using the direct Fick method or thermodilution [1]. Due to the complexity associated with the time and equipment required for direct Fick CO measurement, most PH centers prefer to utilize the thermodilution technique.

New techniques proposed for patient monitoring must be compared with the gold standard (thermodilution, used in our study) rather than with less reliable techniques, such as the indirect Fick method [1, 18], used in Rich *et al.* study [20]. According to the results of the present study, bioreactance does not provide a reliable assessment of CO and cannot be recommended for patient monitoring.

Furthermore, Rich *et al.* evaluated the changes in CO after vasodilator challenge (vasodilator testing using intravenous adenosine) in 36 patients. In 9 of them (25%), there was disagreement regarding the direction of the CO change. The present study also highlighted a weak ability of the Starling™ SV monitor to detect CO worsening during patient follow-up.

In light of recent registry studies, current guidelines recommend a non-invasive follow-up for patients with PH, based on the evaluation of World Health Organization functional class (WHO-FC), 6-minute walk distance (6MWD), and brain natriuretic peptide (BNP) [1, 21, 22]. However, hemodynamic variables, especially those derived from the CO (CI and SVI), are

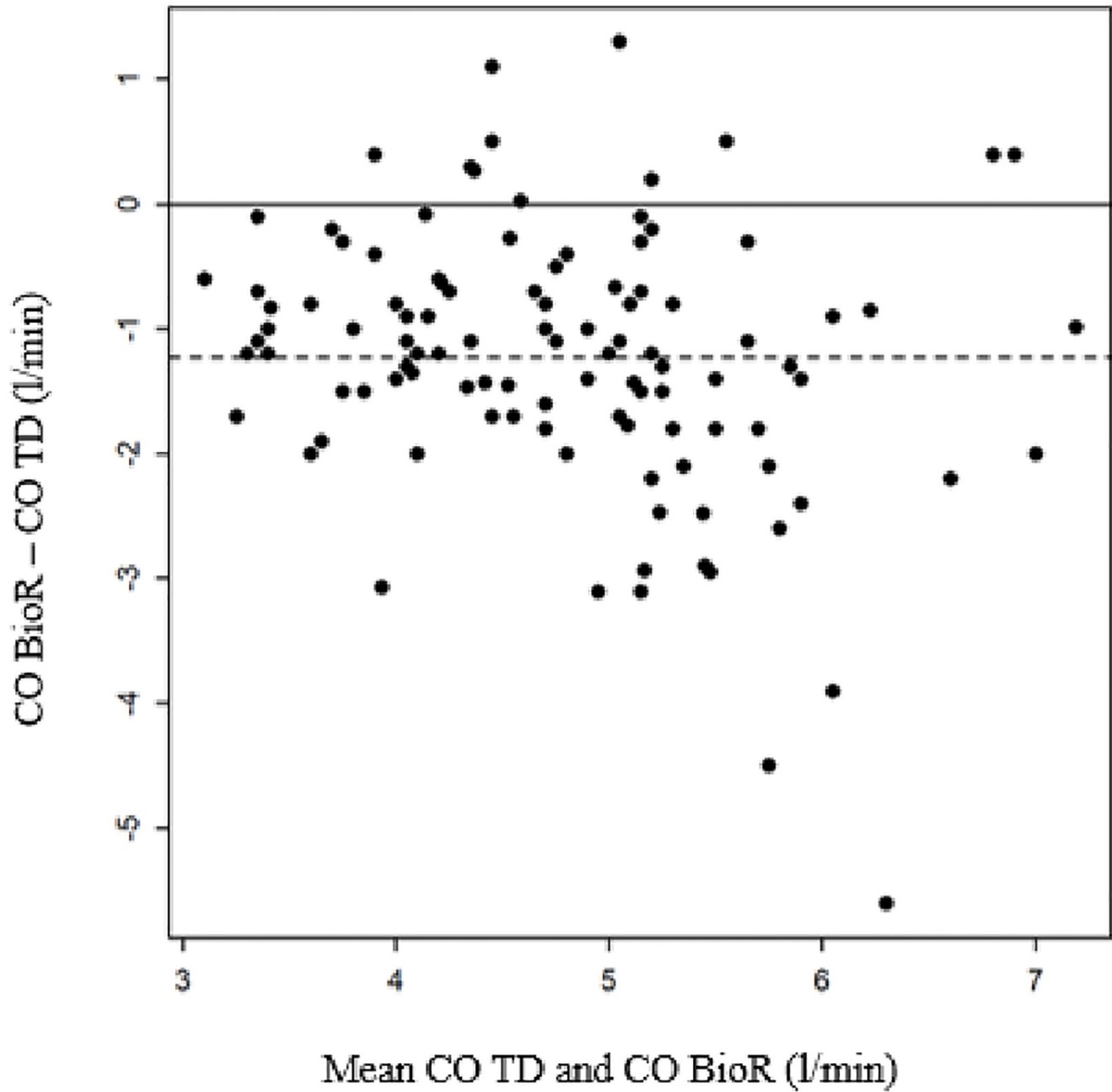

**Fig 3. Bland-Altman plots of CO measurement bioreactance vs thermodilution.** Average deviation = -1.25 l/min [95% CI: -1.48, -1.01].

essential prognostic factors. Their evaluation is recommended not only during the initial management but also at the first reassessment under treatment, in the case of clinical worsening, or when clinical and biological data are discordant [1]. This is why a non-invasive and easily accessible evaluation of CO will be of great interest in practice. Bioreactance is a noninvasive CO measurement technique that is easy and quick to perform, but according to the present results, it was unreliable for the follow-up of patients with PH. A non-invasive and potentially more reliable assessment of the right ventricular (RV) function can also be achieved using

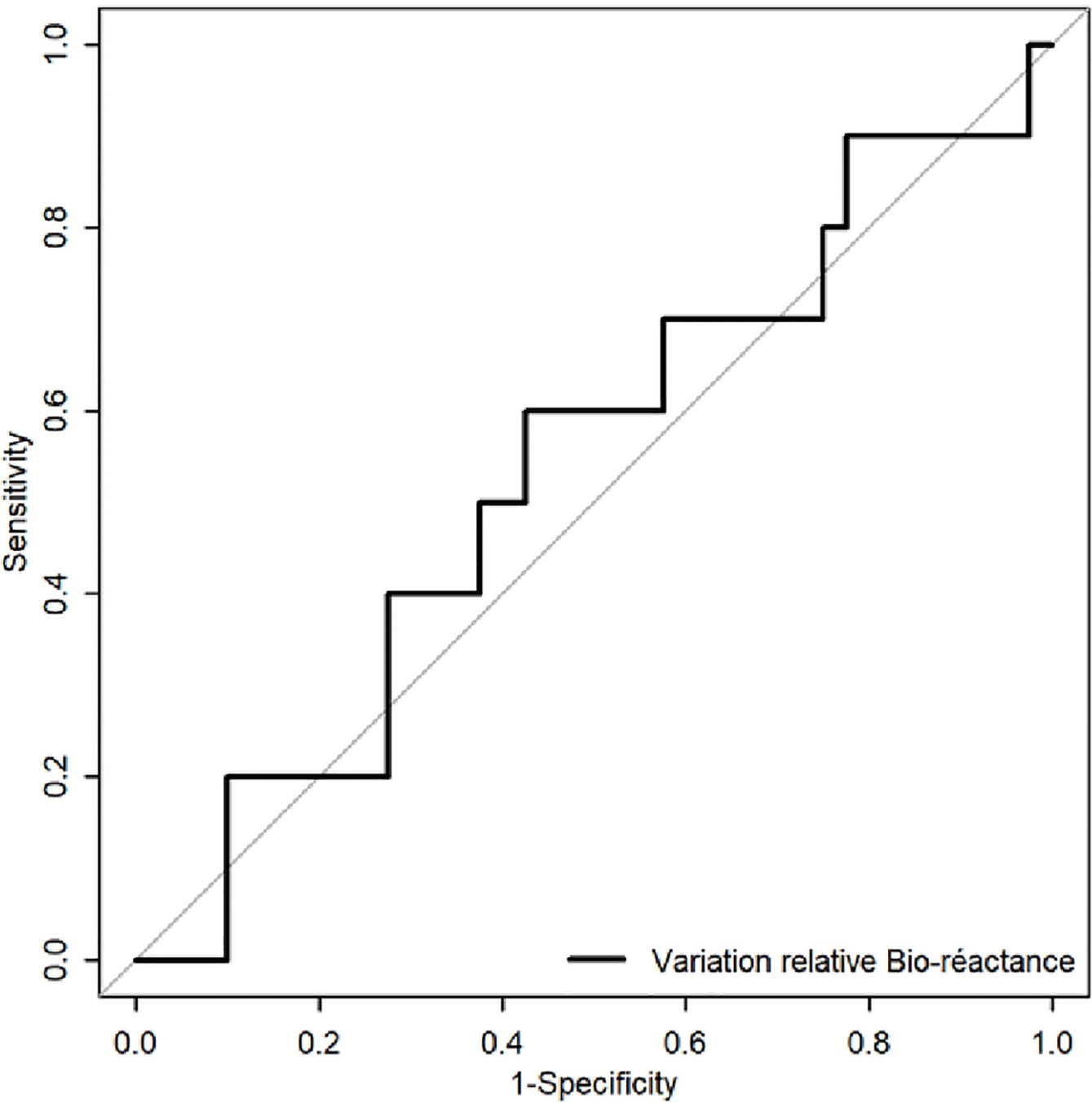

**Fig 4. ROC curve (receiver operating characteristic) for bioreactance ability to detect a variation of CO >15% between V0 and V1 when it exists using the thermodilution technique.** Area under the ROC curve (AUC) = 0.54 [95% CI: 0.33–0.75].

echocardiography, which evaluates parameters such as tricuspid annular plane systolic excursion (TAPSE), TAPSE/systolic pulmonary artery pressure (sPAP), RV fractional area change, RV free-wall strain, tricuspid annulus velocity, and RV ejection fraction [23–26]. Additionally, cardiac magnetic resonance imaging can measure blood flow and CO in the pulmonary artery trunk [27]. Furthermore, conventional planar equilibrium radionuclide angiography allows

the assessment of RV ejection fraction and can be valuable to predict the outcomes of patients with PH [28].

## Limitations

This study had several limitations. First, as per the study protocol, all included patients had mild to moderate PH. Only 10 patients experienced a worsening of CO between V0 and V1, so the ability of the bioreactance monitor to detect variations in CO measurement and distinguish worsening patients was assessed on a small sample. The bioreactance technique was compared to thermodilution, rather than to the indirect Fick method, since thermodilution, according to the guidelines, is the gold standard used in the majority of PH centers, including our own [1]. The accuracy of TD to measure CO has been found to be imprecise in case of catheter migration, tricuspid regurgitation, intra cardiac shunt and can vary during the respiratory cycle [29]. Unfortunately, to date, no CO measurement technique is perfectly accurate. Aware of these pitfalls, we have made effort to avoid potential problems, to the best of our abilities. CO measurements using bioreactance and using thermodilution were not performed simultaneously but within a few minutes from each other. Both techniques were used on immobile patients lying on their back. It is possible that minor variations of CO occurred between the two evaluations. Lastly, it is important to note that this study was conducted at a single center, and the measurements were performed by a single operator.

## Conclusion

The present study reported that during PH, non-invasive measurements of CO using thoracic bioreactance underestimated the values obtained using thermodilution by more than 20%. When applied to the same patients during their follow-up, bioreactance was unable to detect CO worsening. Therefore, the use of bioreactance to monitor patients with PH may not be suitable for clinical practice.

## Author Contributions

**Conceptualization:** Ségolène Turquier, Laure Huot, Fabien Subtil, Vincent Cottin, Jean-François Mornex.

**Data curation:** Ségolène Turquier.

**Formal analysis:** Ségolène Turquier, Fabien Subtil.

**Investigation:** Ségolène Turquier, Medhi Lamkhioued, Julie Traclet, Kais Ahmad, François Lestelle, Louis Chauvelot.

**Methodology:** Ségolène Turquier, Laure Huot, Fabien Subtil, Vincent Cottin, Jean-François Mornex.

**Supervision:** Ségolène Turquier, Vincent Cottin, Jean-François Mornex.

**Validation:** Ségolène Turquier, Laure Huot, Vincent Cottin, Jean-François Mornex.

**Writing – original draft:** Ségolène Turquier.

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
