## [Decision Letter · Decision Letter 0]

9 Jan 2024

PONE-D-23-39899Bioreactance assessment of cardiac output lacks reliability for the follow-up of patients with pulmonary hypertensionPLOS ONE

Dear Dr. turquier,

Thank you for submitting your manuscript to PLOS ONE. After careful consideration, we feel that it has merit but does not fully meet PLOS ONE’s publication criteria as it currently stands. Therefore, we invite you to submit a revised version of the manuscript that addresses the points raised during the review process.

We look forward to receiving your revised manuscript.

Kind regards,

Jörn Karhausen

Academic Editor

PLOS ONE

Journal Requirements:

2. Please amend your manuscript to include your abstract after the title page.

Additional Editor Comments:

In this study Turquier et al. examine the use of Bioreactance assessment for measuring changes of cardiac output in patients with pulmonary hypertension. Here, literature suggests that decline in CO is an important predictor of outcomes in these patients, but current reliance on invasive methods limit the ability to tightly monitor these patients. The main finding of the group is that a) CO measured by bioreactance or thermodilution following right heart catheterization correlate poorly and b) that bioreactance did not adequately detect a significant variation in CO during follow up, when this was found by thermodilution. Together these findings strongly suggest that at the current time Bioreactance assessment cannot substitute more invasive methods to monitor patients CO. Overall, the study is well-conducted, findings are clearly presented and drawn conclusions well-supported by the data. I have only minor concerns:

-Please check language:

e.g. line 38 “To date, only one study was conducted in a cohort of 50 patients, PH.” -> with PH?

Line 94: “Precapillary PH belonged for 75% to group”-> precapillary PH made up 75% of the group?

-Please describe what clinical classification groups 1,4,5 stands for.

-Please clarify abstract, it states that 110 datasets were analyzed. While this is correct, a better representation of the work would be to say that 60 PH patients were included with full datasets available for 50patients on follow up.

Reviewers' comments:

Reviewer's Responses to Questions

**Comments to the Author**

1. Is the manuscript technically sound, and do the data support the conclusions?

Reviewer #1: Yes

2. Has the statistical analysis been performed appropriately and rigorously? 

Reviewer #1: Yes

3. Have the authors made all data underlying the findings in their manuscript fully available?

Reviewer #1: Yes

4. Is the manuscript presented in an intelligible fashion and written in standard English?

Reviewer #1: Yes

5. Review Comments to the Author

Reviewer #1: The investigators report results of a study comparing a noninvasive assessment of cardiac output /stroke volume using a thoracic bio-reactance device to a gold standard of thermodilution derived cardiac output via right heart catheterization.

The main findings of the study was that there was a weak correlation between the noninvasive cardiac output assessment and the thermodilution derived cardiac output, with a negative bias towards lower reported cardiac output using the noninvasive assessment of over 1 L/min.

Importantly, the noninvasive cardiac output assessment was not good at detecting a fall in cardiac output at follow up of 15% with AUC of only 0.54.

The paper is clearly designed and well written. The statistical analysis is sound. I think it is important to publish results of negative studies such as this because it helps focus future research on alternative non-invasive measures of cardiac output and stroke volume.

My only suggestion for revision is to add to the discussion regarding the gold standard of cardiac output assessment. The authors of the article correctly state that thermodilution is typically the recommended means of CO assessment rather than indirect Fick based on ERS guidelines. However the true gold standard is actually direct Fick and there are still major issues with the accuracy of TD in patients with low and high CO and with TR. The authors say they avoided the pitfalls of TD measurement but these are inherent to the method and I am not sure how they can avoid they completely.

The previous study by Rich et. al showed a better correlation between this device and cardiac output by indirect fick but a similar negative bias compared to TD cardiac output. So, in essence, the authors might not really be demonstrating anything different compared to Rich but are comparing the bioreactance CO to a different standard at baseline and follow up. I think the ultimate conclusion - that the device is not adequate for follow up of PH patients - still holds based on their data, but they need to acknowledge that the gold standard they are using is also prone to error.

6. PLOS authors have the option to publish the peer review history of their article (what does this mean?). If published, this will include your full peer review and any attached files.

Reviewer #1: No

---

## [Author Response · Author response to Decision Letter 0]

25 Jan 2024

Dear Pr Karhausen,

Please find appended the revised version of the manuscript entitled “Bioreactance assessment of cardiac output lacks reliability for the follow-up of patients with pulmonary hypertension” (one version with modifications made apparent, and a second clean version), and below a point-by-point reply to comments made.

We hope that this version is acceptable for publication in Plos One, but remain at your disposal if any further modifications are required.

Thank you for your consideration,

Yours sincerely

Ségolène Turquier, on behalf of the co-authors.

Editor comments. 

1. Line 38 “To date, only one study was conducted in a cohort of 50 patients, PH.” -> with PH?

2. Line 94: “Precapillary PH belonged for 75% to group”-> precapillary PH made up 75% of the group?

3. Please describe what clinical classification groups 1,4,5 stands for.

4. Please clarify abstract, it states that 110 datasets were analyzed. While this is correct, a better representation of the work would be to say that 60 PH patients were included with full datasets available for 50patients on follow up.

Authors: 

1. Revised accordingly, line 61 “with PH”

2 and 3. As requested we have changed the sentence, lines 115-122: “In accordance with the last clinical classification of PH, 75% (n=45/60) of cases were attributed to group 1 (pulmonary arterial hypertension) (1). Among these, the distribution was as follows: 31.7% idiopathic; 5.0% heritable; 6.7% associated with drug; 15% associated with connective tissue disease; 3.3% associated with operated congenital heart disease; 1.7% associated with human immunodeficiency virus infection; 8.3% associated with portal hypertension; 3.3% pulmonary veno-occlusive disease. A total of 20% (n=12/60) of patients were in group 4 (PH associated with pulmonary artery obstructions) and 5% (n= 3/60) in group 5 (PH with unclear and/or multifactorial mechanisms).”

4. Following this comment, we have changed the sentence, lines 36, 37: “A total of 60 PH patients were included. All datasets were available at the baseline visit (V0) and 50 of them were usable during the follow-up visit (V1).”

Reviewer comments. 

1. My only suggestion for revision is to add to the discussion regarding the gold standard of cardiac output assessment. … The authors say they avoided the pitfalls of TD measurement but these are inherent to the method and I am not sure how they can avoid they completely.

2. I think the ultimate conclusion - that the device is not adequate for follow up of PH patients - still holds based on their data, but they need to acknowledge that the gold standard they are using is also prone to error

Authors: 

1 And 2. We concur with the reviewer's assessment that no CO measurement technique, whether invasive or non-invasive, is perfectly reliable. Following this comment we have added :

lines 189-191: “Due to the complexity associated with the time and equipment required for direct Fick CO measurement, most PH centers prefer to utilize the thermodilution technique.”

Line 225, 226: “thermodilution, according to the guidelines, is the gold standard used in the majority of PH centers, including our own”

Line 228-230: “Unfortunately, to date, no CO measurement technique is perfectly accurate. Aware of these pitfalls, we have made effort to avoid potential problems, to the best of our abilities.”

In lines 226-228, we discuss the main sources of error in thermodilution, referencing the article authored by AR Opotowsky et al. as a source.

---

## [Editor Report · Decision Letter 1]

30 Jan 2024

Bioreactance assessment of cardiac output lacks reliability for the follow-up of patients with pulmonary hypertension

PONE-D-23-39899R1

Dear Dr. turquier,

We’re pleased to inform you that your manuscript has been judged scientifically suitable for publication and will be formally accepted for publication once it meets all outstanding technical requirements.

Kind regards,

Jörn Karhausen

Academic Editor

PLOS ONE
---

## [Editor Report · Acceptance letter]

6 Feb 2024

PONE-D-23-39899R1 

PLOS ONE

Dear Dr. Turquier, 

I'm pleased to inform you that your manuscript has been deemed suitable for publication in PLOS ONE. Congratulations! Your manuscript is now being handed over to our production team.

Kind regards, 

on behalf of

Dr. Jörn Karhausen 

Academic Editor

PLOS ONE